# Seroprevalence and Associated Risk Factors of Bovine Brucellosis at the Wildlife-Livestock-Human Interface in Rwanda

**DOI:** 10.3390/microorganisms8101553

**Published:** 2020-10-09

**Authors:** Jean Bosco Ntivuguruzwa, Francis Babaman Kolo, Richard Simba Gashururu, Lydia Umurerwa, Charles Byaruhanga, Henriette van Heerden

**Affiliations:** 1Department of Veterinary Tropical Diseases, Faculty of Veterinary Science, University of Pretoria, Pretoria 0110, South Africa; kolofrancis@hotmail.com (F.B.K.); cbyaruhanga27@yahoo.com (C.B.); henriette.vanheerden@up.ac.za (H.v.H.); 2Department of Veterinary Medicine, School of Veterinary Medicine, University of Rwanda, P.O.Box 57 Nyagatare, Rwanda; gasirich@yahoo.fr; 3Department of Animal Resources and Veterinary Services, Rwanda Agriculture and Animal Resources Board, P.O.Box 5016 Kigali, Rwanda; lydiamurerwa88@gmail.com; 4National Agricultural Research Organisation, P.O. Box 259 Entebbe, Uganda

**Keywords:** brucellosis, seroprevalence, potential risk factors, wildlife-livestock-human interface, Rwanda

## Abstract

Bovine brucellosis is endemic in Rwanda; however, little information is available on seroprevalence and risk factors. Therefore, a cross-sectional study was conducted among cattle farmed at the wildlife-livestock-human interface (*n* = 1691) in five districts and one peri-urban district (*n* = 216). Cattle were screened using the Rose Bengal test, then the results were confirmed by indirect enzyme-linked immunesorbent assay. Potential risk factors were determined with a questionnaire and analyzed for their association with seropositivity. In all districts, the animal and herd-level seroprevalence was 7.4% (141/1907) and 28.9% (61/212), respectively, 8.3% (141/1691) and 30.9% (61/198) at the interface, and 0.0% (0/216) in peri-urban areas. Among the potential risk factors, old age (≥5 years), cattle farmed close to wildlife, herds of cattle and small ruminants, history of abortions, and replacement animals were significantly associated with brucellosis (*p* < 0.05). Low awareness of zoonotic brucellosis, assisting calving without biosafety protection, drinking raw milk, and manual milking were each observed in more than 21.7% of cattle keepers whose herds were seropositive. This study confirmed brucellosis endemicity in cattle farmed close to wildlife in Rwanda, suggesting the need to focus control efforts in these areas. Educated farmers with a high awareness of zoonotic brucellosis had low bovine brucellosis seropositivity, which emphasizes the importance of education.

## 1. Introduction

Brucellosis is a neglected and contagious bacterial disease of veterinary and public health importance that remains endemic in developing countries including Rwanda [1]. Brucellosis affects farm and marine animals, wildlife, and humans [2,3]. The disease is caused by bacteria belonging to the genus *Brucella*. Each *Brucella* species has a preferential host, for instance, *Brucella abortus* has a preference for cattle [4], *B. melitensis* for goats and sheep [5], *B. suis* for pigs [6], *B. ovis* for sheep [7], and *B. canis* for dogs [8]. Among these species, *B. melitensis*, *B. abortus*, *B. suis*, *B. canis* cause severe disease in humans [3].

Brucellosis causes tremendous economic losses as a result of abortions, stillbirth, placenta retention, decline in milk yield, and low fertility rates in both females and males [9]. The disease is usually chronic and asymptomatic animals shed infective discharges in the pasture or watering points, and these are important sources of infection for healthy animals [2]. Therefore, regular serological testing of herds would detect infected animals, and lead to control measures to limit the transmission of brucellosis in the herd. Serological testing in cattle involves a combination of more than one serological test [10]. A combination of the Rose Bengal test (RBT) and indirect enzyme-linked immunosorbent assay (i-ELISA) is among the tests of choice due to its high sensitivity and specificity [11,12]. Although two serological tests are recommended for research and diagnosis of brucellosis, only RBT is widely used in most resource-poor countries [13].

In Rwanda, there are few published studies on bovine brucellosis, and individual animal seroprevalence has been found to range from 2.2% in peri-urban areas of Kigali city to 18.9% in the Nyagatare district [14,15,16,17]. The seroprevalence of brucellosis in women with a history of abortion was found to be 6.1% at the Nyagatare district hospital [18] and 25.0% at Huye teaching hospital [19]. Despite the efforts to control brucellosis in Rwanda, the factors responsible for its persistence remain poorly understood. However, the absence of sufficient epidemiological data on the seroprevalence of brucellosis and associated risk factors may impede the design of informed control strategies against brucellosis. The risk factors that have been found to be significantly associated with bovine brucellosis in Rwanda include herd size, breed, and animal age, although that particular study was only conducted in Nyagatare district and the impact of the proximity of livestock to wildlife habitat was not assessed [16].

The herd seroprevalence of brucellosis in cattle farmed at the wildlife-livestock-human interface in East African countries was reported to be 46.7% in Tanzania [20,21], 26.7% [22] and 68.7% [23] in Kenya. An increase of 42.0% in herd prevalence from 2017 [22] to 2019 [23] in the Maasai Mara National Reserve, Kenya, may have resulted from the increased interactions of wildlife and livestock animals due to demographic pressures. Studies in Africa have documented brucellosis seroprevalence in African buffaloes ranging from 7.9% to 20.7% [20,21,24,25]. Strategic control programs are needed for livestock farmed at the interface since the control of the disease in wildlife remains impossible. Although, brucellosis seroprevalence in cattle farmed at the wildlife-livestock-human interface has been documented in neighboring Uganda [26] and Tanzania [21], there is not a single similar study in Rwanda.

The aim of this study was therefore to investigate the seroprevalence of bovine brucellosis and associated risk factors at the wildlife–livestock–human interface. This will contribute to building a database about the occurrence of brucellosis and associated epidemiological factors, and this is essential for providing informed advice to policymakers to improve the control strategies against brucellosis in Rwanda.

## 2. Materials and Methods

### 2.1. Description of the Study Area

This study was conducted in six out of 30 districts of Rwanda; five of which have many dairy cattle at the proximity of national parks, and one district in Kigali city, which has cattle farms in peri-urban areas. The five districts include Nyagatare, Gatsibo, and Kayonza, which border the Akagera National Park in the Eastern Province, Musanze district, which borders Virunga National Park in the Northern Province, and Nyabihu district, which borders the Gishwati-Mukura National Park in the Western Province. The sixth district, Gasabo, located in Kigali city was included to evaluate the brucellosis seropositive status of cattle in peri-urban areas without any proximity to wildlife. Most cattle residing in the Nyagatare, Gatsibo, Kayonza, and Nyabihu districts are crossbreeds, kept in a free grazing system, and most of the farms are fenced with *Euphorbia tirucalli*. On the other hand, the cattle from Gasabo and Musanze districts are kept under a zero-grazing system. Vaccination is not systematic at the national level and several herds that are located in districts of the Eastern Province are in remote areas where access to veterinary services is limited. The climate in the Eastern Province is warmer and drier, characterized by annual average rainfalls ranging between 700–950 mm, and annual average temperatures ranging between 20 °C and 21 °C. The vegetation is grassland with low inclined hills with an average altitude of 1513.5 m. In Northern and Western Provinces, the climate is the coolest and wettest and is characterized by annual rainfalls ranging from 1400–1600 mm and annual average temperatures ranging from 15–17 °C. The topology is mountainous with volcanoes, and the average altitude ranges between 2000 and 3000 m [27]. The Akagera and Virunga National Parks are home to buffaloes (*Syncerus caffer*) [28]. In this study, the wildlife-livestock-human interface was comprised of cattle farms in five districts that border the national parks. Further information on the study area is shown in Figure 1.

### 2.2. Study Design and Sample Size

The study was a cross-sectional design, conducted between May 2018 and September 2019 that applied a multistage cluster sampling strategy to select herds in the selected districts and individual animals within herds. A herd was classified as the sampling unit and this was stratified by districts. The target population was all dairy herds present in the vicinity of national parks or the peri-urban areas of Gasabo district, Kigali city. Cattle aged 1-year-old - and above were selected for this study, and they were categorized as young (1 to 2 years old), adults of medium age (3 to 4 years old), and adults of old age (5 to 13 years old). The dental formula was used to determine the age of animals as previously described [29]. During sampling, a household that had dairy cattle on the same landsite, regardless of the size, was defined as a farm, whereas a farm owned by one or several people, regardless of the size, was considered as one herd. Within each district, households from all areas bordering a national park were randomly selected from sampling frames provided by the district and sector veterinary officers. The study involved blood sampling and herd data collection. The sample size was determined using the formula previously described [30]:$$Sample\;size\;\left(n\right)=\frac{Z^{2\;}P\left(1-P\right)}{d^{2}}$$
where Z^2^ = 1.96 at the 95% confidence level; P is the expected prevalence estimated to be 10% based on a previous study [14] and d is a margin error of 5%. The total sample size per each district was adjusted for clustering using the following equation: N=n (1+ρ (m−1)), where N represents the new sample size, n stands for the original sample size, ρ (=0.2) for the intra-cluster correlation coefficient, and m (=4) represents the number of cattle sampled per herd [30]. The new sample size was 220 cattle per each district. To increase the precision and taking into consideration a large number of cattle in the Nyagatare and Kayonza districts, the sample size was increased by 3 for Nyagatare and 1.5-fold for Kayonza, and this led to 654 and 375 cattle being sampled from the two districts, respectively. The overall sample size was 1907, and these were selected from 212 herds. However, some households consented to participate in the study with the condition of testing all their animals. Therefore, a maximum of four, nine, 15, and more than 15 cattle were selected from 81, 68, 24, and 40 herds, respectively. The sample size at the wildlife-livestock-human interface was 1691 cattle that were selected from 198 herds, while that of peri-urban areas was 216 cattle that were selected from 14 herds.

### 2.3. Questionnaire Design and Data Collection on Individual Cattle and the Herds

Individual data, including the name of the owner, sample identification, age, sex, breed, and location were recorded in a separate list for all selected cattle in the 212 herds. A structured questionnaire comprising of open-ended and closed-ended questions was then administered in a face-to-face interview with one respondent, a cattle keeper/owner of each of the 212 herds to obtain information about potential herd-level risk factors that could be associated with exposure to *Brucella* infection in both cattle and humans. The interviews were conducted in the herd owner’s language (Kinyarwanda) by the primary author or a research assistant who was provided with prior training on all aspects of questionnaire administration in rural and peri-urban areas. The questionnaire was pre-tested in two herds that were not included in the final data set and subsequently adjusted to ensure precision and good flow of the questions and responses. The questionnaire data comprised potential herd-level risk factors, including herd size, herd composition (presence of small ruminants and/or dogs), proximity or history of contact with wildlife, type of grazing system, access to veterinary services, disinfection of pastures and pens, and farmers’ knowledge of the disease, among others. The questionnaire also included questions related to public health to assess the cattle keepers’ knowledge of the zoonotic aspect of bovine brucellosis and predisposing practices in cattle husbandry. The geographical coordinates of each location were recorded using the Global Positioning System (GPS) device (Garmin etrex 10, Lexena, KS, USA) and were then used to generate a map of the study area using ArcGIS (ESRI ArcGIS, version 10.6).

### 2.4. Blood Collection

Samples were taken without causing damage to the animals and respecting their welfare. Blood samples were collected aseptically in a 4-mL plain vacutainer tube from the jugular or tail vein of each selected animal. The vacutainer tubes labelled with each animal identification were transported to the nearest campus of the University of Rwanda where they were stored overnight at room temperature to allow clotting. The following day, sera were collected in a sterile microcentrifuge tube and kept at −20 °C while waiting for serological testing at the Rwanda Agriculture Board, Department of Veterinary Services.

### 2.5. Serological Tests

Animal sera were screened for the presence of *Brucella* antibodies using the Rose Bengal test (RBT, Onderstepoort Biological Products, Pretoria, South Africa) according to the protocol previously described by Alton et al. [31]. Briefly, equal volumes (30 µL) of serum and antigen were mixed for four minutes. A *Brucella* positive and one negative reference samples served as controls. An obvious, clear, and complete agglutination was recorded as a strong (+++) result, while a clear but not complete agglutination was recorded as a medium (++) result. An agglutination that was only visible at the margins was recorded as a weak (+) result. Indirect ELISA was used to confirm RBT positive results in series according to the manufacturer’s instructions (IDvet Diagnostics, Grabels, France). For each test microplate, samples were tested as singles while the positive and negative controls were tested in duplicates. The optical densities (ODs) of samples were determined at 450 nm using an ELISA reader (original multiscan Ex, Thermo Fisher Scientific, Waltham, MA, USA). The sera samples with 120% seropositivity and greater were confirmed positive. In this study, the sera samples showing seropositivity above 119.4% were rounded to 120% and considered positive.

### 2.6. Data Analysis

Individual or herd-level seroprevalence for each district and the entire study were calculated by dividing the total number of animals or herds that were simultaneously positive to RBT and i-ELISA by the total number of animals or herds sampled, multiplied by 100. A herd was considered positive if at least one animal tested positive. Data were recorded and analyzed in Microsoft Excel spreadsheets. Each potential risk factor from the individual- and herd-level data was assessed for significant statistical association with the serological status (considered as a binary outcome: positive or negative), using the Chi-square or Fisher’s Exact tests of association. Variables that were significantly associated with brucellosis seropositivity (*p* < 0.05) at univariate analysis were selected and tested for collinearity using the chi-square test. If a pair of variables was found to be collinear, then only one variable considered to be more biologically associated with brucellosis was considered for multivariable analysis. The screened-in variables were then included in initial multivariable logistic regression models, separately for the individual- and herd-level data. The regression was performed by a generalized linear model (GLM) function, considering a binomial distribution. Subsequently, a stepwise elimination procedure was conducted to arrive at the most adequate model that minimized the Alkaike information criteria (AIC). Univariate and multivariate analyses were performed using R Console version 3.5 (R Core Team, 2017) at a 5% level of significance. The selected model was then subjected to the goodness-of-fit test, by the Hosmer-Lemeshow (χ^2^) test, followed by the determination of odds ratios (OR) for each variable in the final model [32].

### 2.7. Ethical Considerations

This study was approved by the research screening and ethical clearance committee of the College of Agriculture, Animal Sciences and Veterinary Medicine, University of Rwanda (Ref: 026/DRIPGS/2017). Ethical clearance was also obtained from the Institutional Review Board of the College of Medicine and Health Sciences, University of Rwanda (N° 006/CMHS IRB/2018), and the Animal Ethics Committee of the Faculty of Veterinary Science, University of Pretoria, South Africa (V004/2018-2020). Informed verbal consents were obtained from district officials and a consent form was signed by each participant before the commencement of this study.

## 3. Results

### 3.1. Animal and Herd-Level Seroprevalence of Brucellosis in Cattle in Rwanda

The total number of cattle samples analyzed using RBT was 1907, of which 13.6% (260/1907) tested positive. Among these, 260 RBT-positive samples, that is, 45.4% (118/260) were strong positive, 12.3% (32/260) were medium, and 42.3% (110/260) were weak positive. The 260 RBT-positive sera were subsequently analyzed using i-ELISA to confirm the presence of anti-*Brucella* spp. antibodies. Of the 260 RBT-positive samples, 54.2% (141/260) tested positive for brucellosis. The overall true animal-level seroprevalence was 7.4% (141/1907, 95% CI: 6.1, 8.5) using both RBT and i-ELISA, and bovine brucellosis was detected in 83.3% (5/6) of the sampled districts (Table 1). The true animal-level seroprevalence was 8.3% (141/1691, 95% CI: 7.0, 9.7) at the interface, and 0.0% (0/216) in peri-urban district.

The total number of herds analyzed using RBT was 212, of which 49.9% (89/212) tested positive. All the 89 RBT-positive herds were analyzed using i-ELISA to confirm the serological status of brucellosis, and 68.5% (61/89) tested positive for *Brucella* spp. infection. The overall true herd-level seroprevalence was 28.9% (61/212, 95% CI: 22.7, 34.9). Except for Gasabo district in Kigali city, positive herds were recorded in all the other sampled districts (5/6, 83.3%) (Figure 1). The true herd-level seroprevalence recorded at the interface was 30.9% (61/198: 95% CI: 24.4, 34.2), and 0.0% (0/14) in the peri-urban district.

### 3.2. Univariate and Multivariate Analyses of Individual Risk Factors

Univariate analysis of the individual animal risk factors showed that district, animal age, and breed were significantly associated with animal-level seroprevalence (*p* < 0.05). Cattle from the Gatsibo, Nyagatare, and Kayonza districts, which border the Akagera National Park in the Eastern Province showed higher seropositivity than other districts (*p* < 0.05) (Table 1). Among these three districts, Gatsibo and Nyagatare showed significantly higher seropositivity than Kayonza (*p* < 0.05). Older cattle (≥ 5 years) showed the highest seropositivity (9.5%, 74/781) while young animals were least seropositive (3.3%, 9/273). The indigenous breed, “Ankole”, was more exposed (18.0%, 66/367) to *Brucella* spp. compared to the cross-bred (4.8%, 72/1497) and exotic breeds (7.0%, 3/43). Although sex was not significantly associated with brucellosis seropositivity, female cattle were more seropositive (7.5%, 136/1803) than males (4.81%, 5/104) (Table 1).

All of the three variables that were significantly (*p* < 0.05) associated with brucellosis seropositivity in the univariate analysis were included in the final multivariable logistic regression model. Cattle from Gatsibo (OR = 22.2), Nyagatare (OR = 9.7), Kayonza (OR = 7.8), Musanze (OR = 4.2), and Gasabo (OR = 10.0 × 10^−7^) were associated with higher odds of brucellosis seropositivity compared withthe Nyabihu district, although the odds were not statistically significant (*p* > 0.05) for Musanze and Gasabo districts. Cattle of medium age (3 to 4 years old) (OR = 2.4, *p* = 0.03) or older (≥5 years) (OR = 3.0, *p* = 0.005) were associated with significantly higher odds of brucellosis seropositivity (*p* < 0.05) than young cattle (1 to 2 years). The indigenous cattle breed, “Ankole”, was associated with a higher likelihood of seropositivity (OR = 1.8) than the crossbreeds. “Exotic’ breeds were not included in the final logistic regression due to the relatively small number of cattle (43) available for sampling. The Hosmer and Lemeshow goodness of fit test was not statistically significant (χ^2^ = 3.04, *p* = 0.93), showing that the model fitted the data well, with the observed data matching the values expected in theory (Table 2).

### 3.3. Univariate and Multivariate Analyses of Potential Herd Risk Factors

Of the 20 variables considered in the univariate analysis, only 10 showed a significant association (*p* < 0.05) with herd-level seropositivity, and these included herd composition, grazing system, presence of endemic diseases, sharing watering points, history of abortion, good knowledge of bovine brucellosis, access to veterinary services, introduction of new cattle into the herds, and feeding abortion tissues to dogs (Appendix A). Although other herd factors were not significantly associated with brucellosis (*p* > 0.05), high proportions of seropositive animals were observed between levels of variables and these data are available in the Appendix A.

Among the 10 variables that were significantly (*p* < 0.05) associated with brucellosis in the univariate analysis, only six comprised the final multivariate logistic model analysis (Table 3). Herd owners without any level of education (OR = 7.2, *p* <0.05) and those with primary education (OR = 6.7, *p* <0.05) were more likely to have seropositive herds than those with tertiary and secondary education, and the odds were statistically significant. Another important significant predictor for herd-level seropositivity included herd composition with herds that had both cattle and small ruminants being more significantly associated with brucellosis seropositivity (OR = 2.8, *p* < 0.05) compared to herds with cattle only. Good knowledge of animal brucellosis among herd owners was more likely to be associated with brucellosis seropositivity (OR = 5.5; *p* < 0.05). The history of abortions and the introduction of new animals into the herd were also significant predictors (*p* < 0.05) of brucellosis. Cattle reared under free-grazing were associated with higher odds of seropositivity (OR = 1.9) than those under zero-grazing, although the odds were not statistically significant (*p* > 0.05). The Hosmer and Lemeshow goodness of fit test was not statistically significant (χ^2^ = 3.9, *p* = 0.87) showing that the model fitted the data well, with the observed data matching the values expected in theory (Table 3).

### 3.4. Potential Risk Factors Associated with Cattle Keepers Holding Seropositive Herds

Table 4 shows the univariate associations between six risk factors and cattle keepers having seropositive herds. Brucellosis seropositive herds were significantly associated with cattle keepers with insufficient education (*p* < 0.05). Low awareness of zoonotic brucellosis was common in most cattle keepers, 85.9% (182/212) and among them, 26.9% had seropositive animals. Although calving was not significantly associated with herd seroprevalence, most respondents, 76.9% (163/212) assist cattle during parturition without personal protective equipment (PPE), and 31.9% of them had seropositive herds. The number of cattle keepers who drink raw milk was 39.2% (83/212) and of them, 21.7% had seropositive herds. Manual milking was commonly observed in 98.6% of the herds and of them, 28.6% had seropositive herds (Table 4).

There was a significant correlation between awareness of zoonotic brucellosis and boiling milk and between education level and boiling milk (*p* < 0.05). Most of the cattle keepers, 80.0% (24/30) that were aware of brucellosis being zoonotic also boiled their home milk before consumption. Of 182 cattle keepers that were not aware of brucellosis as a zoonotic disease, 42.3% (77/182) drank un-boiled milk. Educated cattle keepers, 79.5% (35/44) were more likely to boil their home milk before consumption compared to uneducated, 58.8% (50/85), and those with primary education, 53.0% (44/83) (Figure 2).

## 4. Discussion

Bovine brucellosis is a contagious bacterial disease of veterinary and public health importance and the disease is endemic in sub-Sahara African countries including Rwanda. This study, which was carried out in six districts, is the first to report on the seroprevalence of brucellosis and associated risk factors in cattle farmed at wildlife-livestock-human interfaces in Rwanda. The findings of the present study confirmed that brucellosis determined with serological tests (RBT and i-ELISA) is endemic in cattle farmed close to the national parks, especially those harboring several buffalos, and the occurrence therein was significantly higher than that in peri-urban areas in the Gasabo district, Kigali city. The overall adjusted animal and herd seroprevalence rates (7.4% and 28.9%) obtained in cattle from six districts in this study as well as the previous rates (9.9–30.2%) obtained in the Nyagatare district of Rwanda using RBT alone [14], the 7.4% rate reported in the Huye district of Rwanda using RBT alone [17], and the rate of 18.9% reported in the Nyagatare district using only RBT [16], confirm that brucellosis is endemic in Rwanda.

Of the 260 (13.6%) sera that were detected as positive for brucellosis using RBT, 118 (45.4%) were strong positive, 32 (12.3%) were medium while 110 (42.3%) were weak positive. Of the 110 RBT-weak positives, 3 (2.7%) were confirmed seropositive using i-ELISA. Most veterinary laboratories in developing countries diagnose brucellosis by detecting only RBT strong positives (complete and clear agglutination) and medium positives (clear agglutination) due to the lack of expertise in detecting weak positives as RBT is a subjective test. Additionally, the confirmation test is not always performed due to either the lack of confirmatory test reagents or the limited number of personnel. Therefore, if the weak positive animals are undetected and then approved for trade, this could contribute to the spread of brucellosis to the naïve herds at the destination.

The animal-level seroprevalence (8.3%) observed in cattle at the wildlife-livestock-human interface is in line with the respective results (8.3% and 9.6%) reported in cattle at the wildlife-livestock-human interface in Zimbabwe using RBT and c-ELISA [33,34], and (9.7%) in Ethiopia using RBT, and i-ELISA [35]. The herd-level seroprevalence of 30.9% observed at the interface in this study is comparable to that obtained in cattle at the wildlife-livestock-human interface in Kenya (26.7%) using i-ELISA [22], and in Ethiopia (32%) using RBT and i-ELISA [35]. On the other hand, our finding was lower compared to the results obtained in cattle farmed at the wildlife-livestock-human interface: in Kenya (68.7%) using i-ELISA [23], and in Zambia (58.1%) using RBT and c-ELISA [36]. This difference was explained by the absence of vaccination programs in the study area in Kenya [23] while in Zambia, the high seroprevalence was associated with abortions and cattle shared grazing pastures and watering points with wildlife [36]. Moreover, the seroprevalence of brucellosis was reported in cattle, buffaloes, and humans at the interface in Tanzania [20,37] and in Zimbabwe [24,25], and this suggests the spillover of brucellosis between these species. The current study together with the above studies confirmed that bovine brucellosis is prevalent in cattle farmed at the wildlife-livestock-human interface, and higher incidences of brucellosis occur in herds with increased interactions between livestock and wildlife.

The animal and herd-level seroprevalence rates observed in this study differed significantly among districts (*p* < 0.05). Cattle from districts that border national parks had higher animal and herd seroprevalences compared to those from peri-urban areas of Gasabo district where no animal was found positive. This difference can be ascribed to the relatively large size of herds, and the free grazing system observed in the Eastern and Western Provinces as compared to the zero-grazing system among cattle farms in Gasabo district, and in which animal health is managed better by the easily accessible veterinary services and readily available animal scientists. Zero-grazing system minimizes contacts between animals and thus reduces the risk of disease transmission. Districts bordering the Akagera National Park in the Eastern Province were more likely to have seropositive cattle (*p* < 0.05) compared to the Musanze and Nyabihu districts that border the Virunga and Gishwati-Mukura National Parks, respectively. In addition, the animal-level seroprevalence recorded in Musanze was high compared to that recorded in Nyahihu. This difference may be attributable to the presence and number of buffaloes within the various national parks. For instance, the Gishwati-Mukura National Park contains only monkeys, chimpanzees, and birds while the Akagera National Park contains many ruminants including buffaloes and the Virunga National Park alongside Musanze has buffaloes. Before the fencing of the Akagera and Virunga National Parks in 2014, cattle grazed and shared watering points with wild herbivores. Although these parks are now fenced, spotted hyenas cross the electric fence from the Akagera National Park to cattle farms (Field observation, 2019). We observed calves wounded around the anus and tail, and these wounds were caused by wild carnivores. These carnivores can move aborted tissue at the wildlife-livestock interface and a recent study isolated *Brucella abortus* and *Brucella suis* from lions and hyenas in the Serengeti National Park, Tanzania [38]. *Brucella abortus* was isolated from 14 dogs in 10 brucellosis positive cattle farms [39] and *Brucella* spp. were isolated from saliva, nasal discharges, and urine of dogs feeding on aborted tissue [40,41] and urine has been incriminated in the transmission of canine brucellosis [42]. In addition, occasional transmission of brucellosis through bites has been reported [43]. Thus, the movement of these carnivores feeding on aborted tissue and live calves and goats in both the park and cattle farms may play a role in the transmission of bovine brucellosis and other zoonotic diseases between the wildlife and livestock and vice-versa. Elsewhere in Africa, significantly higher brucellosis seroprevalences were reported in cattle in areas close to wildlife habitat compared to areas far from the home range of wildlife, i.e., in Uganda [26] and Tanzania [37,44]. The significantly higher occurrence of brucellosis observed in districts that border national parks can be attributed to previous interactions between wildlife and livestock, and indirect interactions by carnivores and rodents. It is therefore worth further investigating the occurrence of brucellosis in buffaloes and other wild animals in Rwanda.

This study showed that the age of the cattle was a significant predictor of brucellosis seropositivity, with the medium adult age category (3 to 4 years) and the old cattle (≥5 years) being more affected (OR = 5, *p* = 0.005) than young animals. This finding is in agreement with other studies carried out in Rwanda [14,16], and in Uganda [45]. Animals that are kept for a longer period in the herds have more chances of exposure and acquiring brucellosis, and this translates into increased brucellosis seropositivity with increasing age. It has also been reported that *Brucella* spp. have a tropism for reproductive organs of mature female animals, and the sex hormones and erythritol produced are responsible for the survival and multiplication of *Brucella* species [46]; this contributes to the overall higher seropositivity in sexually mature females.

In this study, herds in which cattle grazed together with small ruminants were significantly more likely to be seropositive than cattle-only herds, which is consistent with similar studies that found that mixing cattle and small ruminants was a significant predictor of brucellosis [47,48]. This suggests that small ruminants may play a role in the maintenance and persistence of brucellosis in cattle in Rwanda since the former are not vaccinated. This also indicates that there may be co-infection with *B. melitensis* and *B. abortus* in the same herd which is consistent with a recent study in South Africa that isolated both species in slaughtered cattle [49].

A history of abortions was a significant predictor for herd-level seroprevalence, and this is in agreement with previous reports from Uganda [26,45], and Tanzania [50]. Furthermore, this study also revealed that 98.6% of respondents did not dispose of abortuses properly and birth sites were not disinfected, which is consistent with a previous report in Nyagatare district [16]. Therefore, it is likely that there will be a continuous circulation of *Brucella* pathogens within and between herds. Various reproductive disorders that are associated with brucellosis have been reported in the cattle industry in Rwanda, including higher incidences of abortions, retained placenta, infertility of unknown origin, and longer calving intervals [51]. Such abortions can cause tremendous financial losses and wherever they occur in the herd, massive screening of the herd against brucellosis is very important and positive animals should be immediately slaughtered to stop the spread.

Our findings revealed that uneducated and less educated cattle keepers were significantly associated with higher herd-level seropositivity than herds whose owners had attained secondary and/or tertiary education. These findings are in agreement with those of Assenga et al. [44], who reported lower *Brucella* infection exposure in the herds of educated livestock farmers [44]. Illiterate or less educated farmers are likely to be less informed or to adopt slowly to innovations, and this may be matched with poor management practices such as the hygiene of cattle and their environment, and weaker implementation of recommended control measures such as the restriction of animal movements and vaccination. Indeed, we found that among the 26 cattle herders who vaccinated their animals, 88.5% (23/26) were educated. Nevertheless, education and learning are processes, and owners with less or no education can be helped, for example, through regular consultation with professionals in animal science. Furthermore, 78.8% (167/212) of cattle keepers had good knowledge of bovine brucellosis, which is known as “Amakore” in the Kinyarwanda language. The farmers knew that the disease is characterized by abortions and hygromas in the patellofemoral joint. However, herds belonging to such farmers had higher odds (5.5 times) of brucellosis seropositivity than herds with no awareness, which indicates that despite the knowledge, there is negligence in implementing recommended control measures such as restriction of movement and removal of seropositive animals from the herd. In contrast, several other studies reported poor knowledge of brucellosis among several cattle keepers [16,35], and this was associated with an increase of *Brucella* infection in herds.

The majority of cattle keepers (85.9%) did not know that brucellosis affects humans. It is therefore not surprising that 60.9% of cattle keepers mentioned that they drank boiled milk not to avoid brucellosis, but to prevent diarrhea or tuberculosis. Also, boiling milk was significantly associated with awareness of zoonotic brucellosis, and with education level. The low awareness of zoonotic brucellosis is further reflected in the observation that a high proportion (76.9%) of cattle keepers that assisted calving without wearing protective equipment or clothing—and given that manual milking was observed in almost all (98.8%) of the herds—this constitutes a high risk for cattle keepers. In congruence with our findings, low awareness of zoonotic brucellosis was also reported in more than 92.0% of cattle keepers in Ethiopia [35], of which, most farmers did not regard exposure to abortion tissues, drinking and eating raw animal products as risk factors. In this study, boiling milk was significantly associated with awareness of zoonotic brucellosis, and with education level (*p* < 0.05). Within cattle keepers, the increase in awareness of zoonotic brucellosis or in their education level influences an increase in the number of cattle keepers boiling milk before home consumption. This finding indicates that continuing education of cattle keepers and other exposed groups on the epidemiology of zoonotic brucellosis and other zoonotic diseases should contribute significantly to preventing zoonotic diseases in humans.

## 5. Conclusions

This study confirmed that brucellosis is endemic in cattle farmed at the wildlife-livestock-human interface and found that the history of abortions and introduction of new animals into herds are the major predictors of brucellosis. Therefore, aborting cattle, and cattle for replacement should be quarantined, tested, and the positives slaughtered. The interface should be more targeted by control programs such as vaccinations, testing and slaughter, and the requirement of an annual brucellosis-free certificate for national and international trade. Most cattle keepers had low awareness of zoonotic brucellosis and this was exemplified by them assisting calving without PPE and improper disposal of abortion tissues. This awareness should be raised among all stakeholders through education campaigns on zoonotic brucellosis. The One Health concept of involving veterinarians, environmentalists, and physicians could efficiently minimize zoonotic brucellosis, and the control of animal brucellosis would prevent the disease in humans since there is no vaccine for the latter. Further studies on brucellosis seroprevalence in wildlife, carnivores, and humans living at the interface are worthy of investigation in Rwanda.

## Figures and Tables

**Figure 1 microorganisms-08-01553-f001:**
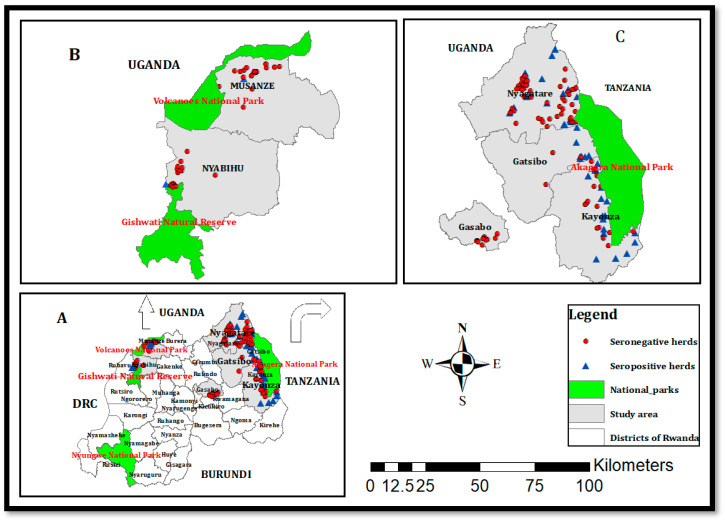
Maps of (**A**) Rwanda with different districts, (**B**) the Musanze and Nyabihu districts border the Virunga and Gishwati national parks, respectively, and (**C**) the Nyagatare, Gatsibo, and Kayonza districts border Akagera National Park, and Gasabo is an urban district with peri-urban areas. Red circles and blue triangles indicate seronegative and seropositive herds found in this study.

**Figure 2 microorganisms-08-01553-f002:**
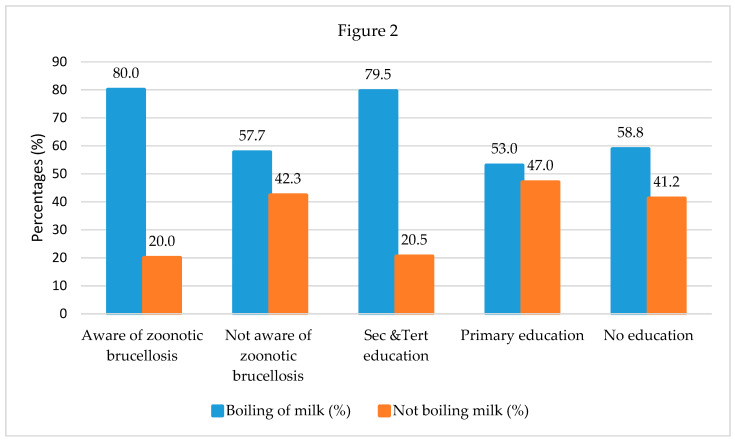
Pairwise correlation between awareness of zoonotic brucellosis, education level, and drinking raw milk in cattle keepers in Rwanda.

**Table 1 microorganisms-08-01553-t001:** Results of descriptive and univariate analysis between potential individual animal risk factors and the serological status of brucellosis in cattle sampled at the wildlife-livestock-human interface in Rwanda.

Variables	Categories	Positive to RBT			Positive to i-ELISA			Positive to RBT & i-ELISA		
		NT	Total n^+^ (%)	*p*-Value	NT	Positive n^+^ (%)	*p*-Value	NT	Positive n^+^ (%)	*p*-Value
Districts	Gasabo	216	20 (9.3)	<0.001 ^a^	20	0 (0.0)	<0.001 ^a^	216	0 (0.0)	<0.001 ^a^
	Gatsibo	226	41 (18.1)		41	40 (97.6)		226	40 (17.7)	
	Kayonza	375	83 (22.1)		83	38 (45.8)		375	38 (10.1)	
	Musanze	215	13 (6.1)		13	7 (53.9)		215	7 (3.3)	
	Nyabihu	220	11 (5.0)		11	2 (18.2)		220	2 (0.9)	
	Nyagatare	655	92 (14.1)		92	54 (58.7)		655	54 (8.2)	
Age	Young (1–2 years)	273	26 (9.5)	0.013 ^a^	26	9 (34.6)	0.098	273	9 (3.3)	0.002 ^a^
	Medium (3–4 years)	853	108 (12.7)		108	58 (53.7)		853	58 (6.8)	
	Older (≥5 years)	781	126 (16.1)		126	74 (58.7)		781	74 (9.5)	
Sex	Female	1803	249 (13.8)	0.43	249	136 (54.6)	0.56	1803	136 (7.5)	0.4
	Male	104	11 (10.6)		11	5 (45.5)		104	5 (4.8)	
Breeds	Exotic breeds *	43	6 (14.0)	<0.001 ^a^	6	3 (50.0)	1	43	3 (7.0)	0.004 ^a^
	Cross	1497	155 (10.4)		155	72 (46.5)		1534	99 (4.8)	
	Ankole	367	99 (27.0)		99	66 (66.7)		330	39 (18.0)	

RBT, Rose Bengal Test; i-ELISA, indirect enzyme-linked immunosorbent assay; NT: number of cattle tested; n^+^: number of positive animals; Exotic breeds * included Friesian and Jersey. The total number of samples analyzed using RBT was 1907, of which 260 tested positive. The 260 RBT-positive samples were subsequently analyzed using i-ELISA for confirmation of the brucellosis status. ^a^: proportions are significantly different (*p* < 0.05).

**Table 2 microorganisms-08-01553-t002:** Results of multivariable logistic regression between animal-level risk factors and serological status of brucellosis in cattle sampled at the wildlife-livestock-human interface in Rwanda.

Variables	Category	Odds Ratios	95% CI	*p*-Value
Districts	Nyabihu ^a^			
	Gasabo	10.0 × 10^−7^	0.00–inf.	0.975
	Gatsibo	22.2	5.3–93.3	<0.001 ^b^
	Kayonza	7.8	1.7–35.7	0.008 ^b^
	Musanze	4.2	0.9–20.6	0.075
	Nyagatare	9.7	2.3–40.1	0.002 ^b^
Age	Young ^a^			
	Medium	2.4	1.1–5.1	0.025 ^b^
	Older	3.0	1.4–6.3	0.005 ^b^
Breeds	Crossbreed ^a^			
	Ankole	1.8	1.0–3.3	0.067

^a^ Reference categories for comparing serological status amongst cattle. ^b^
*p* < 0.05: significant difference in serological status as compared to the reference level for each variable. Hosmer and Lemeshow χ^2^ = 3.5, df = 8, *p*-value = 0.9.

**Table 3 microorganisms-08-01553-t003:** Results of multivariable logistic regression between potential herd risk factors and the serological status of brucellosis in cattle farmed at the wildlife-livestock-human interface in Rwanda.

Variables	Category	Odds Ratio	95% CI	*p*-Value
Education category	Tertiary ^b^			
	Primary	6.7	1.9–23.3	0.003 ^a^
	None	7.2	2.1–24.4	0.001 ^a^
Herd composition	Cattle only ^b^			
	Cattle and SR	2.8	1.1–6.7	0.024 ^a^
	Cattle and dogs	1.4	0.6–3.4	0.458
Grazing system	Zero-grazing ^b^			
	Free grazing	1.9	0.8–4.5	0.144
Brucellosis knowledge	No ^b^			
	Yes	5.5	1.7–18.1	0.005 ^a^
History of abortions	No ^b^			
	Yes	2.5	1.2–5.1	0.014 ^a^
New introduction	No ^b^			
	Yes	2.7	1.3–5.9	0.011 ^a^

^b^ Reference categories for comparing serological status amongst cattle. ^a^
*p* < 0.05: Significant difference in serological status as compared to the reference level for each variable. Hosmer and Lemeshow χ^2^ = 3.9, df = 8, *p*-value = 0.9.

**Table 4 microorganisms-08-01553-t004:** Univariate associations between public health risk factors and herd brucellosis seropositivity among cattle keepers residing at the wildlife-livestock-human interface in Rwanda.

Variables	Categories	Sample Size	No. of Responses (%)	Odds Ratio	95% CI	*p*-Value
Education level	Tert.& sec.	44	4 (9.1)	-	-	0.002 ^a^
	Primary	83	25 (30.1)			
	None	85	32 (37.7)			
Zoonotic brucellosis	Yes	30	12 (40.0)	0.6	0.3–1.3	0.21
	No	182	49 (26.9)			
Boiling milk	Yes	129	43 (33.3)	0.6	0.3–1.0	0.094
	No	83	18 (21.7)			
Assisting calving	Yes	163	51 (31.3)	1.7	0.8–3.8	0.194
	No	49	10 (20.4)			
Using PPE	Yes	0	0 (0.0)	-	-	1
	No	163	51 (30.1)			
Milking method	Manual	210	60 (28.6)	2.5	0.1–98.1	0.493
	Machine	2	1 (50.0)			

No.: number; tert.& sec.: tertiary and secondary; ^a^
*p* < 0.05: significant difference in the frequency of responses; CI: confidence interval.

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
