# Peer review of "Seroprevalence and Associated Risk Factors of Bovine Brucellosis at the Wildlife-Livestock-Human Interface in Rwanda"

_microorganisms, 2020, doi:10.3390/microorganisms8101553_

Round 1

Reviewer 1 Report

Dear Authors,

the topic you have dealt with is very interesting not only for the experts in the sector but also for further raising awareness among the Veterinary Public Health sector regarding the epidemiological status of bovine brucellosis in Rwanda and in the world.

I believe that further studies and insights are needed, as the authors have indicated, to support these data of theirs but I am confident that this article after publication can trigger some interest, exploring further several aspects concerning the wide field of bovine brucellosis.

In my opinion, I found the document well-structured and detailed in the introduction and methods sections. In contrast, I found the description of the results heavy and structured in an unclear way for readers. In this regard, even the tables and the figures are not formatted adequately and sometimes not contained synthetic information.

Therefore, I recommend that a detailed revision is performed. Here are my comments and suggestions in more detail. I kindly request that the authors specifically address each of my comments in their revision note.

Comments and Suggestions

Since the manuscript was not formatted with line numbers, I tried to indicate the different corrections by referring to the page and the section or subsection.

Page 1

Please change the font size of “Abstract”.

The sentence “The animal and herd-level seroprevalences were: 7.4% (141/1907) and 28.9% (61/212) in all the districts;” is not very clear. I suggest you modify it as follows: “In all the districts, the animal and herd-level seroprevalences were 7.4% (141/1907) and 28.9% (61/212), respectively;”

Please change the style of “Keywords” in bold.

Page 2-3

The final sentence in the Subsection 2.1 “All animal vaccines are subsided and controlled by Rwanda Agriculture Board (RAB) who distributes them to district veterinarians. The policy of vaccination against brucellosis consists of administering RB 51 to calves on demand by herders upon payment of RWF 500 (approximately $ 0.6) per dose.” Is not consistent in this context. Please remove.

Page 3

The paragraph "Akagera National Park [...] and different species of birds [29]." it should be summarized, as too many details are reported about the flora and fauna of the study context.

Figure 1 is very well done and is important. I only suggest changing the size of the legend.

Page 4

Please change the "Study design and sample size" subsection number from 2.1 to 2.2. Change the numbers in the next subsection in ascending order.

Study design and samples size”: Why did the authors use the dental formula to determine the age of the animals? Is there no bovine registry in Rwanda? At least one reason should be given.

Questionnaire design and data collection on individual cattle and the herds”: The study carried out using this way is very interesting and useful for the evaluation of risk factors. It would be advisable to share the Questionnaire used with readers and attach it as "Supplementary materials".

Page 5

Please change the "Blood collection" subsection number from 2.1 to 2.4. Please integrate the first sentence of this subsection, stating that the samples were taken without causing damage to the animals, respecting their welfare.

Change the numbers in the next subsection in ascending order.

Page 6

Please modify the number of the section “Results” from 2 to 3.

Please change the “Animal and herd-level seroprevalence of brucellosis in cattle in Rwanda” subsection number from 2.1 to 3.1. Change the numbers in the next subsection in ascending order.

Table no. 1 is not clearly legible because its formatting is not in line with the instructions for the authors of the Journal. Please check and modify.

Page 7

Please modify the number of the subsection “Univariate and multivariate analyses of potential herd risk factors” from 2.1 to 3.3.

Table no. 2 is not formatted is not in line with the instructions for the authors of the Journal. Also, the caption is unclear and not homogeneous in font size. Please check and modify.

Page 8

The caption of table n. 3 is not clear. Please check and modify.

Please modify the number of the subsection “Potential risk factors associated with cattle keepers holding seropositive herds” from 2.1 to 3.4.

Page 9

The paragraph below the table n. 4 should be written in the normal style and not in caption style. In addition, what the Authors mean for “Table 0”?

Figure n. 2 should be made easier to read. Probably, the type of graph (e.g. histograms) should be changed as it does not have to show the trend of data over time. Also, in the figure please change "parcentages" to percentages.

The caption should be written without the colon after "Figure 2" and in the normal style.

Please modify the number of the section “Discussion” from 2 to 4.

Page 10

The sentence at the end of the second paragraph "This limitation could result ....." is of little use to a scientific audience. Please remove it.

Page 11

In the second paragraph, please correct "Brucellas pp" with Brucella spp. ".

In the last paragraph, replace (2016) with [44], in line with the Journal's "Instructions for Authors".

Page 12

Please modify the number of the section “Conclusions” from 2 to 5.

In this section, I suggest changing one health to "One Health"

References

The references must be wide changed, since they were not written following the “instructions for authors" of the Journal.

Reviewer 2 Report

The manuscript by Jean B. Ntivuguruzwa and colleagues describes a brucellosis seroprevalence study in Rwanda and the analysis of potential risk factors for Brucella spp. infection. The manuscript is clearly written and logically organized. The content is technically sound, and overall the research is well described. The conclusions are supported by the analysis of the data presented. However, "Discussion" section sometimes is difficult to follow and some sentences should be revised for clarification.

I would like to point out some comments:

Results: 

  • Serological data refers only to cattle or also to small ruminants? 
  • How may herds were included in this study?
  • The authors refer that 89 out of 212 herds were recorded as RBT positive. It is assumed that they were also iELISA positive. This should be clarified in the text. 
  • In page 6, line 226, the authors refer that "68.5% (61/89) were positive for Brucella spp." This study included bacteriological analysis? If yes, this should be clarified in the text; if not, references should be added about this data.

Table 1: Tables should be self-explanatory. Please refer to the total of tested animals.

Discussion: 

Some sentences are confused or repeated. Just as examples:

- page 9, lines 316-323.

- page 10, lines 332-347.

Minor comments

Page 1, lines 38: "Brucella species affecting mainly livestock, wildlife, and humans..." is already mentioned in the previous line. Please rephrase the sentence. 

Page 2, line 47: Please delete "ruminants" and replace by "cattle".

Page 4, lines 120-121. In the sentence "Red circles 121 and blue triangles indicate seronegative and seropositive herds obtained in this study." the authors should refer to Figure 1.

Page 5, lines 190-207: Please add references regarding statistical methods.

Page 7, line 260-262 and page 8, line 283: Please correct the formatting.
